# How Do Aromatic Nitro Compounds React with Nucleophiles? Theoretical Description Using Aromaticity, Nucleophilicity and Electrophilicity Indices

**DOI:** 10.3390/molecules25204819

**Published:** 2020-10-20

**Authors:** Kacper Błaziak, Witold Danikiewicz, Mieczysław Mąkosza

**Affiliations:** 1Faculty of Chemistry, University of Warsaw, 01-224 Warsaw, Poland; 2Biological and Chemical Research Center, University of Warsaw, 01-224 Warsaw, Poland; 3Institute of Organic Chemistry, Polish Academy of Sciences, Kasprzaka 44/52, 01-224 Warsaw, Poland; witold.danikiewicz@icho.edu.pl (W.D.); mmakosza@icho.edu.pl (M.M.)

**Keywords:** nucleophilic aromatic substitution, reaction mechanism, DFT

## Abstract

In this study, we present a complete description of the addition of a model nucleophile to the nitroaromatic ring in positions occupied either by hydrogen (the first step of the S_N_Ar-H reaction) or a leaving group (S_N_Ar-X reaction) using theoretical parameters including aromaticity (HOMA), electrophilicity and nucleophilicity indices. It was shown both experimentally and by our calculations, including kinetic isotope effect modeling, that the addition of a nucleophile to the electron-deficient aromatic ring is the rate limiting step of both S_N_Ar-X and S_N_Ar-H reactions when the fast transformation of σ^H^-adduct into the products is possible due to the specific reaction conditions, so this is the most important step of the entire reaction. The results described in this paper are helpful for better understanding of the subtle factors controlling the reaction direction and rate.

## 1. Introduction

Nucleophilic aromatic substitution of halogens in halonitroarenes is one of the fundamental processes of organic chemistry widely used for industrial and laboratory organic synthesis [1,2,3]. The reaction proceeds via the addition of nucleophiles to the aromatic rings at positions occupied by halogen X to form σ^X^-adducts followed by spontaneous departure of X^−^ anions to give the substitution products. The addition is connected with dearomatization of the rings, whereas formation of the product by departure of X^−^ results in rearomatization, thus, as a rule, the addition is the slow, rate limiting step of the reaction. This mechanism, formulated by J.F. Bunnett [4,5], was confirmed in numerous mechanistic studies and presently is generally accepted [6]. It has been also confirmed by numerous quantum chemical calculations of the reaction pathways between halonitrobenzenes and nucleophiles [7,8,9,10,11,12,13,14,15,16,17,18]. These calculations showed that depending on the structure of the reactants—nitroarenes and nucleophiles—the σ^X^-adduct can either be an intermediate, i.e., is located in a local energy minimum on the potential energy surface (PES) of the reaction, or a transition state [19].

The nitro group in halonitrobenzenes activates three positions of the ring (two *ortho* and one *para* position) towards nucleophilic addition, thus the nucleophiles can add also at positions occupied by hydrogen. Indeed, there are a few old reports of reactions that proceed via addition of nucleophiles to *p*- and *o*-chloronitrobenzenes at positions occupied by hydrogen and further conversions of the produced σ^H^-adducts, such as von Richter reaction [20] or formation of benzisoxazoles [21]. Some years ago, it was shown that the addition of nucleophiles to halonitroarenes at positions occupied by hydrogen proceeds faster than at those occupied by halogens and that there are a few general ways of conversion of the initially formed σ^H^-adducts into products of nucleophilic substitution of hydrogen S_N_Ar-H, such as oxidation [22,23], vicarious nucleophilic substitution (VNS) [24,25], etc. (Scheme 1). In fact, oxidative nucleophilic substitution of hydrogen, ONSH, vicarious nucleophilic substitution of hydrogen, VNS, and some other variants of S_N_Ar-H are presently well recognized and widely used processes in organic synthesis [3,26,27,28,29,30,31,32]. On the basis of these results, it was concluded that classical S_N_Ar-X of halogens is a secondary process preceded by fast and reversible formation of the σ^H^-adducts [33].

However, even in recent publications on the mechanism of S_N_Ar reactions this situation was not taken into account [3,34]. Moreover, in all quantum chemical calculations of the energy profiles of the addition of nucleophiles to halonitrobenzenes only the addition at positions occupied by halogens has been considered [7,8,9,10,11,12,13,14,15,16,17,18,34]. It is really surprising, because calculations of a reaction between two reactants should look for the pathways that proceed via transition states (TS) of the lowest free energies, thus they have ignored faster addition at positions occupied by hydrogen.

Previously we have presented results of DFT calculations of the reactions between a model nucleophile: carbanion of chloromethyl phenyl sulfone **1** and three model nitroarenes: nitrobenzene **2**, *p*-fluoro- and *p*-chloronitrobenzenes **3** and **4** for the gas phase and DMF solutions. These calculations have shown that the addition at positions occupied by hydrogen proceeds via TS of lower free energy than at those occupied by halogens (Figure 1); thus, they are in agreement with the experimental results. On this basis, a real, corrected mechanism of nucleophilic aromatic substitution has been formulated [35,36,37]. The model nucleophile (chloromethyl phenyl sulfone anion) has been chosen as an example, which, due to its nature, is much less affected by the solvent’s effects contrary to the other most common protic or polar nucleophiles [35].

The aim of this paper is to present a full mechanistic picture of nucleophilic aromatic substitution in halonitroarenes and nitrobenzene embracing such subtle features as changing aromaticity of the nitroarenes as well as electrophilicity of nitroarenes and nucleophilicity of the carbanion in the addition process, kinetic isotope effects and effects of substituents on the rates of the addition on the basis of DFT calculation and experimental results.

## 2. Results and Discussion

As the model nucleophile in our calculations, we have used the carbanion of chloromethyl phenyl sulfone **1** that was used in experimental mechanistic studies of S_N_Ar-H reaction. It was also used in the previous calculations of the energy profiles of the reaction with nitrobenzene **2**, *p*-fluoronitrobenzene **3** and *p*-chloronitrobenzene **4** (Scheme 2).

### 2.1. Aromaticity

Nucleophilic addition to the nitroaromatic rings, regardless of positions of the addition, is connected with dearomatization. In fact, the intermediate adducts, particularly σ^H^-adducts, are relatively stable entities and have the structure of cyclohexadienenitronate anions [38]. It is therefore of great interest to follow the change of the aromaticity in the course of the reaction.

The results of experimental studies, particularly rates of the reactions, can clarify some key elements of the mechanism. In the case of S_N_Ar-X and S_N_Ar-H, they can be interpreted in terms of changes of aromaticity in the reaction course. Nonaromatic structure of σ^H^-adduct to *p*-substituted nitrobenzenes was directly determined by observation of its ^1^H NMR spectra [38].

These experimental results provide information about the key steps of the reaction. Changes of aromaticity in the course of nucleophilic addition can be followed by quantum chemical calculations. In order to follow electronic reorganization during this process, calculations based upon the Harmonic Oscillator Model of Aromaticity (HOMA) [39] were performed. When the nucleophile approaches the nitroarene ring and the distance between them decreases, the excess of negative charge affects the shape of the ring. Figure 2 shows the HOMA aromaticity index as a function of the progress of the reaction between our model nucleophile, the anion of chloromethyl phenyl sulfone **1** and **2**, **3** and **4**. In all cases the addition to the ring at positions occupied by hydrogen: *ortho*- and *para*- of **2** and *ortho*- of **3** and **4**, size of the ring increases and it loses planarity. The circumference of the ring increases in all these cases by about 2% (see Appendix A). It is therefore evident that the loss of aromaticity in the case of the σ^H^-adducts is the largest, in agreement with experiments. It is a different situation in the case where the nucleophile adds to *p*-halonitrobenzenes at the position occupied by halogen. The addition at the position occupied by fluorine in **3** results in formation of the σ^F^-adduct, located in a minimum of the free energy on the reaction profile. This adduct possesses higher aromaticity than σ^H^-adduct and circumference of the ring increases by ca. 1.7%, perhaps because the electron withdrawing fluorine atom accepts a partially negative charge of the ring.

The addition of **1** at position *para* of **4** occupied by chlorine results in the formation of the structure identical to the σ^Cl^ adduct, however, it is not located at the local minimum of Δ*G* on the reaction path but at the saddle point, thus it is the transition state (Figure 3). This phenomenon has been discussed in recent reports by Fernández et al. [19] and earlier in the case of electrophilic reactions by Gwaltney et al. [40]. Additionally, the observation that the σ^Cl^-adduct is the transition state structure was omitted in a very innovative work by Ormazábal-Toledo et al. [16] in which the authors introduce the calculation methodology used by us in the next parts of the present work. The lowest aromaticity of the system is observed after the transition state when the rearomatization process starts with the departure of the chloride anion. The negative charge is removed from the ring with the leaving group and the nitrobenzene ring relaxes into the symmetric planar shape. The circumference of the ring increases by 1.9% when the σ^Cl^-adduct is formed as the transition state.

This part of the investigation shows clearly that the S_N_Ar reaction substitution of fluoride proceeds fastest because during the reaction there is the smallest aromaticity loss, so the atomic system has the best possibility to recover aromaticity. The replacement of chloride is slower because the larger loss of aromaticity is observed during the reaction. The σ^Cl^-adduct is not a stable intermediate product, which is why the S_N_Ar reaction between *p*-chloronitrobenzene and the nucleophile is irreversible. These observations based on the aromaticity are in agreement with the kinetics. Finally, the addition at positions occupied by hydrogen smoothly leads to the highest aromaticity loss and the formation of stable intermediate σ^H^-adducts, which can be transformed into the final S_N_Ar-H product by several ways under the appropriate reaction conditions [24,25,26,27,28,41,42,43,44,45].

### 2.2. Electrophilicity and Nucleophilicity

The concepts of nucleophilicity and electrophilicity are an inseparable part of nucleophilic substitution processes. Over the years, many attempts to determine their absolute and relative values, both experimentally [46,47,48,49,50,51,52,53,54,55,56] and theoretically [16,18,57,58,59,60], have been reported. In the particular case of nucleophilic aromatic substitution of halogens (S_N_Ar-X) and hydrogen (S_N_Ar-H), in which the initial step is nucleophilic addition to the electron-deficient ring, electrophilicity is the factor deciding the rate of addition.

Thus, the effect of substituents on the rate of the addition can be considered as the effect on electrophilicity of the ring. Although, as a rule, the addition is the rate limiting step of S_N_Ar reaction, the effect of substituents on the rate of substitution of halogen in halonitrobenzenes containing various substituents cannot be considered as a measure of electrophilicity of the ring, because it depends on the kind of halogens and, particularly, because it is a secondary reaction preceded by fast and reversible addition at positions occupied by hydrogen. Thus, real electrophilic activity of substituted nitroarenes was determined in relation to the rate of the addition reaction at positions occupied by hydrogen. For technical reasons, relative rates of the addition of the model nucleophile **1** were determined in relation to the rate of addition at position *ortho* of nitrobenzene. These experimentally determined electrophilicities of *p*-substituted nitrobenzenes are in good correlation with the calculated electrophilicity indices ω^+^ of these nitroarenes (Table 1, Figure 4). These data describe electrophilicity of the nitroarenes before the reaction. Changes of electrophilicity of nitroarenes and nucleophilicity of **1** along the reaction path can be tracked only by computation [60]. Following the reaction coordinates, calculations of nucleophilicity and electrophilicity were performed (for more details please consult Appendix A) for selected parts of the atomic system^16^, according to models shown in Figure 5.

Based on conceptual density functional theory [62,63], the electrophilicity and nucleophilicity indices for the addition of the selected nucleophile in the *ortho* position of nitrobenzene derivatives were calculated, Figure 6.

It was proven by the calculated activation barriers [35] (Figure 1) that the rate of σ^H^-adduct formation is strictly related to substituent effects in the aromatic ring. The electron-withdrawing effect of the halogen in the *para* position contributes to the activation of the *ortho* position towards nucleophilic attack. In this case the results obtained for *p*-fluoro-, *p*-chloro- and unsubstituted nitrobenzene are very similar. The upper left diagram in Figure 6 shows the calculated electrophilicity of permanent group (PG). Each nitrobenzene derivative has different electrophilic character, thus the potential to react with the nucleophile decreases accordingly: *p*-chloro- > *p*-fluoro- > unsubstituted nitrobenzene. The electrophilicity of PG fell rapidly en route to the formation of σ^H^-adducts and it slightly increased in the nucleophile atomic system (lower left diagram, Figure 6). The upper right diagram shows that the calculated nucleophilicity of the permanent group increases during the addition in the *ortho* position. Nucleophilicity values of the permanent group are mainly affected by the type of the nucleophile and its character as a donor of extra electron density. This causes the differences of nucleophilicity between nucleophile acceptors to be less pronounced.

On the other hand, the same factors in the form of electrophilicity and nucleophilicity indices for the S_N_Ar reaction were calculated. The results suggest that the electrophilicity of the permanent group in the case of *para* substitution is very similar to the loss of aromaticity and the nucleophilicity trend of PG is reversed, Figure 7.

For *p*-chloronitrobenzene, the electrophilicity decreases until the transition state structure is formed; after the chloride anion detaches, the potential to accept an electron from the environment increases. For *p*-fluoronitrobenzene and nitrobenzene, the trends of the factors are very similar to those obtained for the reaction in the *ortho* position. The reactions in the *para* position of *p*-fluoronitrobenzene and nitrobenzene lead to the formation of σ^F^-adduct and σ^H^-adduct, respectively. That observation is also consistent with the aromaticity changes and the properties of the formed σ-adducts, where the σ^F^-adduct and the σ^H^-adduct are stable intermediates, and the σ^Cl^-adduct is a transition state structure. The substitution of the chlorine atom as a first order reaction is also visible in the case of nucleophilicity change trends of the leaving group, where this parameter increases for the chloride ion after its departure from the nitrobenzene ring. All of the presented trends for S_N_Ar-X and S_N_Ar-H reactions seem to reproduce reliably the nucleophilicity and electrophilicity changes along the electron reorganization path during the bond formation-bond breaking processes between the nucleophile and the nitrobenzene derivatives.

### 2.3. Kinetic Isotope Effect

Studies of kinetic isotope effects, KIEs, are an important tool for clarification of mechanisms of organic reactions. Since differences of rates of reactions in which isotopes are involved are functions of their mass differences, KIEs of hydrogen ^1^H vs. ^2^D are the most often studied. Nevertheless, on the basis of the KIEs of fluorine ^18^F vs. ^19^F, it was shown that in S_N_Ar of fluorine in 2,4-dinitrofluorobenzene in the reaction with piperidine, the addition or the elimination can be the rate limiting step depending on the solvent [63,64]. On the other hand, determination of KIEs of hydrogen H vs. D was of great importance for determination (elaboration) of the mechanism of S_N_Ar-H reactions [65,66]. Thus, it was shown that the addition of the model carbanion **1** at positions occupied by hydrogen of 4-bromo-2-deuteronitrobenzene proceeds somewhat slower than at identically activated positions occupied by deuterium, KIE ≈ 0.8. Calculations of the energy profiles for this reaction in the gas phase by three DFT methods gave the results between 0.62 and 0.82, which are in a good agreement with the experimental data. These results show that calculation of KIE can be quite useful in studying reaction mechanisms, especially that it is much simpler compared to the experiment.

## 3. Conclusions

In conclusion, the detailed mechanistic picture of both variants: S_N_Ar-X and S_N_Ar-H of nucleophilic aromatic substitution pathways in halonitroarenes was fully described by the reactivity indices and kinetic isotope effect. The latter shows unequivocally that the addition of the nucleophile to the aromatic ring is the rate-determining step of all variants of S_N_Ar-X mechanism and of S_N_Ar-H, in specific conditions where the σ^H^-adduct can be easily transformed to the products. Our results showed that there is a significant difference between reaction mechanisms of the substitution of chlorine and fluorine atoms. The first reaction is a single step process, in which the σ^Cl^-adduct is a transition state. In contrast, in the case of the substitution of the fluorine atom, the σ^F^-adduct is an intermediate product, which is always the case in the S_N_Ar-H reactions. Our results show that the formation of σ^H^-adducts is preferred and, providing that they can undergo fast further transformations, the S_N_Ar-H reaction is dominating over the S_N_Ar-X. Fluoronitrobenzene is a special case because both σ^H^- and σ^F^-adducts are intermediate products so the relative rate of the substitution of fluorine and hydrogen depends on the reaction conditions. Finally, we have shown that reactivity indices, when used correctly, are a very useful tool for describing mechanisms of the reactions between aromatic nitro compounds and nucleophiles.

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
