# Peer review of "How Do Aromatic Nitro Compounds React with Nucleophiles? Theoretical Description Using Aromaticity, Nucleophilicity and Electrophilicity Indices"

_molecules, 2020, doi:10.3390/molecules25204819_

Round 1

Reviewer 1 Report

The manuscript describes the improved mechanistic picture of SNAr-X and SNAr-H processes of nucleophilic aromatic substitution in halogenonitro- and nitrobenzenes. Сompared with previously published results of DFT calculations of SNAr reaction (ref.34) authors completed the mechanism description by essential features such as aromaticity (HOMA), electrophilicity and nucleophilicity of reactants. Performed calculations are useful for understanding the SNAr mechanism. The manuscript can be accepted after the following corrections:

1) The choice of a model nucleophile for the mechanism investigation should be briefly justified.

2) The calculations show that p-chloronitrobezene reacts with nucleophile without formation of σ-complex as intermediate, and the loss of aromaticity is realized in the transition state. But according to the very recent publication (Org.Biomol.Chem. 2020, 18, 4238) a “hidden” Meisenheimer complex may exist in such reactions. Did the authors observe a small shoulder on the IRC profile of the SNAr reaction between p-chloronitrobenzene and the model nucleophile?

3) It is not entirely clear how the curves of changes in electrophilicity and nucleophilicity are obtained (fig. 6,7), i.e. how many points were used to construct the curves.

4) In SI the geometry of different TS is given, but there are no comments in the text about these calculations. It would be useful to give the images of TS1-TS6 for different 4-X-nitrobenzenes

5) The number of imaginary frequencies, characteristic frequencies for Transition States and thermochemistry data should be given in SI for optimized structures.

6) The table of contents in SI should be corrected (p.S5 should be removed).

7) p.4, entries 99, 100: should be “ortho- and para- of 2 and ortho- of 3 and 4,” instead of “ortho- and para- of 2 and ortho- of 3 and 3,”?

Author Response

Response to Reviewer 1 Comments

The manuscript describes the improved mechanistic picture of SNAr-X and SNAr-H processes of nucleophilic aromatic substitution in halogenonitro- and nitrobenzenes. Сompared with previously published results of DFT calculations of SNAr reaction (ref.34) authors completed the mechanism description by essential features such as aromaticity (HOMA), electrophilicity and nucleophilicity of reactants. Performed calculations are useful for understanding the SNAr mechanism. The manuscript can be accepted after the following corrections:

1) The choice of a model nucleophile for the mechanism investigation should be briefly justified.

Authors response:

A brief justification has been added to the main text in the lines no. 62-64. The model nucleophile has been chosen as an example that is much less affected by solvation effects than e.g. protic neutral NH3 or other -OH or methoxylate anions. In our previous paper from 2016 (ref. 35) we have shown a very good agreement between the experiment and theory both in gas-phase approximation and using the implicit solvent model. In addition, our model nucleophile is a very good mimic candidate for QQ calculation performed even in the gas-phase due to the fact that the synthetic experiments are carried out in dipolar aprotic solvent, which limits the number of potential weak (H-bond type) interactions that may influence on the mechanism.

2) The calculations show that p-chloronitrobezene reacts with nucleophile without formation of σ-complex as intermediate, and the loss of aromaticity is realized in the transition state. But according to the very recent publication (Org.Biomol.Chem. 2020, 18, 4238) a “hidden” Meisenheimer complex may exist in such reactions. Did the authors observe a small shoulder on the IRC profile of the SNAr reaction between p-chloronitrobenzene and the model nucleophile?

Authors response:

During the IRC calculations, which have been performed to confirm the authenticity of each transition state, no visible minimum for SNAr-Cl reaction was observed. In our opinion, this situation, as described in the cited paper (Org.Biomol.Chem. 2020, 18, 4238) is possible to observe (using DFT computational methods) only for specific nucleophiles, like for those with balanced-charged nitrogen-based nucleophiles, where the elimination of leaving group (HCl) is accompanied with N-H bond cleavage. The cited manuscript is very interesting from a theoretical point of view, and it has been included in the reference list. However, it seems to be another example where the faster and primary reaction in which at the first step the nucleophile adds to the carbon occupied by the hydrogen atom in the aromatic ring is not even computationally considered. 

3) It is not entirely clear how the curves of changes in electrophilicity and nucleophilicity are obtained (fig. 6,7), i.e. how many points were used to construct the curves.

Authors response:

Thank you for this very important comment. The description on how the elecrophilicity and nuclephilicity curves have been prepared has been added to the S1 chapter in SI. The computational strategy has been adopted based on publication authored by R. Ormazábal-Toledo et.al (J. Org. Chem., 2013, 78, 1091-1097.).

 4) In SI the geometry of different TS is given, but there are no comments in the text about these calculations. It would be useful to give the images of TS1-TS6 for different 4-X-nitrobenzenes

Authors response:

The additional comment about the transition state geometry analysis and the information on which transition state geometry has been chosen for IRC calculation, used for plotting the nuclephilicity/electrphilicity and HOMA profiles have been added to the S1 chapter in SI documentation.

5) The number of imaginary frequencies, characteristic frequencies for Transition States and thermochemistry data should be given in SI for optimized structures.

Authors response:

All the missing data has been added to the SI.

6) The table of contents in SI should be corrected (p.S5 should be removed).

Authors response:

The correction has been introduced.

7) p.4, entries 99, 100: should be “ortho- and para- of 2 and ortho- of 3 and 4,” instead of “ortho- and para- of 2 and ortho- of 3 and 3,”?

Authors response:

The correction has been introduced.

Reviewer 2 Report

The authors present a very interesting theoretical contribution moving from the results of their landmark work published in 2016 (ref.34). The most relevant point lies in the detailed, proper description of the role of nucleophilic attack at a CH position in the activated aromatic ring, a reversible step that has long been undervalued in mechanistic studies on nucleophilic aromatic substitution reactions, mostly focused on the overall substitution process involving a leaving group departure from a CX substituted site.  In this manuscript emphasis is placed on a semiquantitative description based on aromaticity, electrophilicity and nucleophilicity indices. This investigation allows to underline the major factors affecting reactivity, placing an accent on the aromaticity parameter along the course of the reaction.

In my view this paper represents a remarkable contribution allowing a valuable insight into the detailed mechanism of this important class of reactions and should be published as is after some thorough check on the English language that will probably be done at editorial stage.

In forthcoming work it may be desirable to obtain information into the role that a polar or even protic solvent may exert on the nucleophilic aromatic substitution reaction involving ionic species.

Author Response

Response to Reviewer 2 Comments

The authors present a very interesting theoretical contribution moving from the results of their landmark work published in 2016 (ref.34). The most relevant point lies in the detailed, proper description of the role of nucleophilic attack at a CH position in the activated aromatic ring, a reversible step that has long been undervalued in mechanistic studies on nucleophilic aromatic substitution reactions, mostly focused on the overall substitution process involving a leaving group departure from a CX substituted site.  In this manuscript emphasis is placed on a semiquantitative description based on aromaticity, electrophilicity and nucleophilicity indices. This investigation allows to underline the major factors affecting reactivity, placing an accent on the aromaticity parameter along the course of the reaction.

In my view this paper represents a remarkable contribution allowing a valuable insight into the detailed mechanism of this important class of reactions and should be published as is after some thorough check on the English language that will probably be done at editorial stage.

In forthcoming work it may be desirable to obtain information into the role that a polar or even protic solvent may exert on the nucleophilic aromatic substitution reaction involving ionic species.

Authors response:

We would like to express our thanks for these positive comments on our manuscript. The influence of the solvent molecules' surroundings on the kinetic and thermodynamic parameters of the SNAr reaction is a very important but also very challenging task from a computational point of view. Especially, the solvation of ionic or neutral, -NH or –OH type of nucleophiles by protic, polar solvents. This kind of solvent-molecule interaction leads to the formation of very dense and intimate effects in the form of a hydrogen bond net, which may affect the overall mechanism. One of our current projects is about to employ the dynamic computational methodologies (CPMD) in which the solvent will be treated explicitly in a periodic manner and test its influence on the reactant molecules and the shape of the minimum energy pathway en route to the SNAr products.